# Genetic Aspects of Myelodysplastic/Myeloproliferative Neoplasms

**DOI:** 10.3390/cancers13092120

**Published:** 2021-04-27

**Authors:** Laura Palomo, Pamela Acha, Francesc Solé

**Affiliations:** 1MDS Group, Institut de Recerca Contra la Leucèmia Josep Carreras, ICO-Hospital Germans Trias i Pujol, Universitat Autònoma de Barcelona, 08916 Badalona, Spain; lpalomo@carrerasresearch.org (L.P.); pacha@carrerasresearch.org (P.A.); 2Experimental Hematology, Vall d’Hebron Institute of Oncology (VHIO), Vall d’Hebron Barcelona Hospital Campus, Universitat Autònoma de Barcelona, 08035 Barcelona, Spain

**Keywords:** myelodysplastic/myeloproliferative neoplasms, cytogenetics, molecular landscape, gene mutations

## Abstract

**Simple Summary:**

Myelodysplastic/myeloproliferative neoplasms (MDS/MPN) are clonal myeloid neoplasms characterized, at the time of their presentation, by the simultaneous presence of both myelodysplastic and myeloproliferative features. In MDS/MPN, the karyotype is often normal but mutations in genes that are common across myeloid neoplasms can be detected in a high proportion of cases by targeted sequencing. In this review, we intend to summarize the main genetic findings across all MDS/MPN overlap syndromes and discuss their relevance in the management of patients.

**Abstract:**

Myelodysplastic/myeloproliferative neoplasms (MDS/MPN) are myeloid neoplasms characterized by the presentation of overlapping features from both myelodysplastic syndromes and myeloproliferative neoplasms. Although the classification of MDS/MPN relies largely on clinical features and peripheral blood and bone marrow morphology, studies have demonstrated that a large proportion of patients (~90%) with this disease harbor somatic mutations in a group of genes that are common across myeloid neoplasms. These mutations play a role in the clinical heterogeneity of these diseases and their clinical evolution. Nevertheless, none of them is specific to MDS/MPN and current diagnostic criteria do not include molecular data. Even when such alterations can be helpful for differential diagnosis, they should not be used alone as proof of neoplasia because some of these mutations may also occur in healthy older people. Here, we intend to review the main genetic findings across all MDS/MPN overlap syndromes and discuss their relevance in the management of the patients.

## 1. Introduction

Myelodysplastic/myeloproliferative neoplasms (MDS/MPN) constitute a heterogeneous group of clonal myeloid malignancies with clinical, laboratory, morphologic and genetic features that overlap with myelodysplastic syndromes (MDS) and myeloproliferative neoplasms (MPN). According to the 2017 World Health Organization (WHO) classification, this category currently includes four adult subtypes: chronic myelomonocytic leukemia (CMML), *BCR-ABL1*–negative atypical chronic myeloid leukemia (aCML), MDS/MPN with ring sideroblasts and thrombocytosis (MDS/MPN-RS-T), MDS/MPN-unclassifiable (MDS/MPN-U), and one pediatric entity: juvenile myelomonocytic leukemia (JMML) [1].

MDS/MPN are usually characterized by a hypercellular bone marrow (BM) with increased proliferation in one or more of the myeloid lineages which is also accompanied by dysplastic features (as a result of increased programmed cell death). Simultaneously, cytopenia may also be present. The blast percentage in the BM and peripheral blood (PB) should be <20% [1].

Depending on the subtype, conventional cytogenetics allows the identification of chromosomal abnormalities in 10–50% of the cases, while around 90% of patients present with somatic mutations in myeloid-related genes [2,3,4]. In this review, we intend to summarize the main genetic findings across all MDS/MPN overlap syndromes and discuss their relevance in the management of patients.

## 2. Diagnostic Criteria of MDS/MPN

As previously mentioned, MDS/MPN represents a heterogeneous group of myeloid malignancies that share clinicopathological features with both MDS and MPN. According to the WHO criteria, diagnosis is primarily based on morphological and laboratory findings, as well as exclusion of specific genetic abnormalities [1].

The most common and most well characterized MDS/MPN subtype is CMML, which is characterized by sustained (≥3 months) PB monocytosis (≥1 × 10^9^/L; monocytes ≥ 10%) and BM dysplasia [1]. Its incidence is estimated in four cases per 100,000 people per year [5]. Median age at diagnosis is 72 years and it is an infrequent disease in young adults [6,7]. Clinical course is highly variable, with a median overall survival (OS) that ranges between 12–24 months and 15–30% probability of progression to acute myeloid leukemia (AML) [6]. CMML was initially considered as an MDS subtype by the French–American–British (FAB) classification, which subdivided this entity based on leukocyte count into myelodysplastic (MD-CMML, <13 × 10^9^/L) and myeloproliferative (MP-CMML, ≥13 × 10^9^/L) variants [8]. In 2001, when the WHO assigned CMML to the overlap MDS/MPN group, two categories (CMML-1 and CMML-2) were distinguished according to BM or PB blast percentage [9]. In this case, the percentage of blasts represents the sum of monoblasts, promonocytes and myeloblasts. Both classifications hold prognostic value, since patients with MP-CMML or CMML-2 have shorter OS and a higher risk of AML transformation [10]. Years later, Schuler et al. proposed a refined categorization where CMML-1 subtype was divided into two groups [11]. Based on all these, current WHO classification recognizes three CMML categories: CMML-0 (<2% blasts in PB and <5% blasts in BM), CMML-1 (2% to 4% blasts in PB and/or 5% to 9% blasts in BM) and CMML-2 (5% to 19% blasts in PB, 10% to 19% in BM, and/or when any Auer rods are present), but also recommends the separation of CMML into MD/MP-CMML, since this can guide the therapeutic approach [1].

Atypical CML is defined largely by morphologic criteria including leukocytosis, dysplastic neutrophils and their precursors. Cytogenetic and molecular studies should be negative for Philadelphia chromosome and *BCR-ABL* fusion gene [1]. The exact incidence of aCML is unknown, but it is estimated in <2 cases for every 100 cases of *BCR-ABL1*-positive CML [12]. Overall, aCML is generally associated with a very poor prognosis and a median OS of 10–20 months [4,13,14].

MDS/MPN-RS-T was a provisional entity until the 2017 WHO classification update, and it is characterized by the presence of thrombocytosis (≥450 × 10^9^/L), large atypical megakaryocytes, anemia and ring sideroblasts accounting for ≥15% of erythroblasts [1]. *SF3B1* mutation is reported in ~90% of patients [15]. In contrast to MDS-RS, the diagnosis of MDS/MPN-RS-T cannot be established if *SF3B1* mutation is accompanied by 5–<15% ring sideroblasts. It represents the subtype associated with the best prognosis among overlap syndromes, with a median OS of around 6 years [16].

MDS/MPN-U is the most heterogeneous and the least well-characterized entity, including patients that do not meet other MDS/MPN diagnostic criteria. Median OS is reported in 15–25 months and leukemia-free survival in 19 months [17,18,19]. Clinical characteristics and the natural history of patients with MDS/MPN-U are not well established, due to the heterogeneity of the patients, although poor prognosis among patients with MDS/MPN-U is reported in several studies [17,18,20].

Finally, JMML, the childhood counterpart of CMML, is a rare heterogeneous myeloid neoplasm that shares many clinical and molecular aspects of CMML, and is currently considered a bona fide RASopathy syndrome. It is the only pediatric-onset neoplasm within MDS/MPN and is characterized by excessive proliferation of granulocytic and monocytic lineages [21]. Age at diagnosis ranges from 1 month to early adolescence, but 75% of cases occur in children aged < 3 years [1,21]. Splenomegaly is present almost in all cases. The clinical course varies widely, thus, appropriate clinical management ranges from watchful observation to early allogenic hematopoietic stem cell transplantation (HSCT) [22].

## 3. Cytogenetic Abnormalities in MDS/MPN

In general, cytogenetic abnormalities and somatic copy number alterations (CNAs) are uncommon and unspecific across all MDS/MPN subtypes (Table 1, Figure 1A), considering that the same alterations are also found in other myeloid malignancies. In most cases, prognosis is not well defined for specific alterations.

The majority of CMML patients have a normal karyotype; however, around 25–30% present with clonal cytogenetic abnormalities. Common alterations include trisomy 8 (+8), loss of Y chromosome (−Y), abnormalities of chromosome 7 (chr7), trisomy 21 (+21) and complex karyotypes (≥3 cytogenetic abnormalities; CK) [23]. Trisomy 8 is commonly found in isolation and is detected in 6–7% of CMML patients [23,24]. Chr7 abnormalities, which mainly constitute monosomy 7 (−7) and 7q deletion (del(7q)), are reported in 2–9% of CMML cases [23,24]. These abnormalities are also present at different frequencies in other myeloid neoplasms such as MDS, MPN and AML. Finally, −Y is reported in 4–6% of CMML patients [23,24,25], although its impact is a matter of debate because, even when it has been described in several neoplasms, it is also found in healthy elderly men [28,29]. To date, three different CMML-specific cytogenetic risk classification systems have been proposed, which stratify patients in groups that differ in their OS and risk of AML progression [23,24,25] (Table 2). According to these, normal karyotypes and isolated −Y are associated with favorable outcomes. In contrast, chr7 abnormalities, CK and monosomal karyotypes (defined by the presence of two monosomies or one monosomy + ≥1 structural abnormality) are associated with a poor outcome, while the prognostic impact of +8 remains controversial.

In the case of aCML, few patient cohorts with cytogenetic data have been described until now [3,4,13,24], with the largest series consisting of 65 and 71 patients, respectively [4,24]. According to these two studies, approximately 43% of patients present with cytogenetic abnormalities, +8 and chr7 alterations being the most common, with reported frequencies of approximately 17% and 7%, respectively. CK are seen in 4–8% of the cases. In contrast, only 10–17% of MDS/MPN-RS-T patients have an abnormal karyotype. Commonly detected alterations include +8 and -Y, while other chromosomal abnormalities, as well as CK, are rare (0–4%) [4,15,26]. Among all MDS/MPN overlap syndromes, MDS/MPN-U is the subtype with the highest frequency of chromosome instability, with near 50% of altered karyotypes. Trisomy 8 (mostly found as a sole abnormality) is the most frequent alteration (15–25%), followed by chr7 alterations (12%) and CK (12%) [4,20,27]. Other less common abnormalities include del(12p), +9 and del(20q) [20,27]. Overall, the presence of cytogenetic abnormalities is generally associated with an inferior OS in all adult MDS/MPN subtypes, except aCML [4]. This impact seems to be especially strong in MDS/MPN-RS-T, where abnormal karyotypes are rare but, if detected, confer a very poor outcome [4,26] (Table 2).

Cytogenetic studies of JMML show a normal karyotype in approximately 65–80% of cases [21,39,40]. Monosomy 7 is the most frequent alteration, reported in 9–25% of the cases. Other aberrations (such as del(7q) and +8) are reported in 10% of cases [21,39]. It is to note that −7 is most often seen in *KRAS*-mutated cases [22].

## 4. Other Chromosomal Abnormalities

As previously mentioned, the karyotype is often normal across all MDS/MPN subtypes [1]. Studies using single nucleotide polymorphism arrays (SNP-A) or other techniques that allow the detection of cryptic CNAs and copy number neutral loss of heterozygosity (LOH) are limited. To date, most of these studies report MDS/MPN cases (mainly CMML) within heterogeneous cohorts including other myeloid malignancies [41,42,43]. Overall, SNP-A allows the detection of chromosomal alterations in 75% of MDS/MPN patients compared to 30–40% by conventional cytogenetics [41,42,43,44,45,46].

Two SNP-A studies performed in large cohorts of CMML patients with normal karyotype reported 40–65% of abnormalities (CNAs + LOH) in these patients [45,46]. According to these studies, CNAs are detected in one third of patients but are highly heterogeneous, with very few recurrent alterations, including gains in 21q22 and losses in 4q24 and 12p13.2. In contrast, large interstitial LOH regions are detected in 25–35% of cases, recurrently affect 4q, 7q and 11q, and are often accompanied by the presence of homozygous mutations in *TET2*, *EZH2* and *CBL*, respectively. Prognostic impact of these abnormalities remains unclear.

Besides CMML, LOH were also reported in 38% MDS/MPN-U, especially in 11q23.3 were *CBL* gene is located [44]. Similarly, Jankowska et al. described that LOH of chromosome 4q (where *TET2* gene is located) was frequent in CMML and in secondary AML arising from these cases; however, it was absent in refractory anemia with ring sideroblasts and thrombocytosis (current MDS/MPN-RS-T) and aCML patients [47].

Overall, very few cryptic alterations are seen in patients with MDS/MPN, either by SNP-A or sequencing techniques, and these are highly heterogeneous and not specific to any of the subtypes [4]. Thus, even when chromosomal microarray testing is included as a suggested test by the European LeukaemiaNet 2013 and by the Spanish Group of MDS for the diagnosis of primary MDS, it has not been recommended for clinical work-up of myeloid malignancies by the WHO 2017, nor by the NCCN 2017 guidelines [48].

## 5. Functional Pathways Affected in MDS/MPN

The development of next generation sequencing (NGS) techniques has helped to define the molecular landscape of MDS/MPN. More than 90% of these patients harbor somatic mutations in a group of genes that is common across the spectrum of myeloid neoplasms [4]. Pediatric entity JMML, considered a RASopathy, is primarily characterized by germline and somatic mutations in genes involved in the RAS pathway [38]. In contrast, the spectrum of gene mutations in adult MDS/MPN is much more heterogeneous, with driver genes affecting specific cellular processes that can be categorized according to their function. Mutations recurrently affect epigenetic regulators, splicing factors, genes involved in signaling pathways, transcription factors and cohesin complex components [2,3,4,26,37] (Figure 1B and Figure 2). The acquisition of mutations in these patients occurs in a multi-step manner, as reported in both myeloid and lymphoid neoplasms [49]. Many cases probably arise from previous asymptomatic clonal hematopoiesis, and thus founder driver mutations are frequently found in epigenetic regulators and splicing factors. Secondary acquired driver mutations commonly affect transcription factors and signal transduction genes, which sometimes drive disease progression to AML, along with cell-intrinsic and -extrinsic factors. However, gene mutation frequencies and clonal evolution patterns differ among the four adult MDS/MPN subtypes [4] (Figure 1B and Figure 3).

### 5.1. Epigenetic Regulators

Mutations in epigenetic regulators are very common across the spectrum of myeloid malignancies and constitute the most frequent type of somatic mutations detected in MDS/MPN, seen in up to 75% of adult overlap syndromes [4]. They can be divided into DNA methylation enzymes (*TET2*, *DNMT3A* and, less frequently, *IDH2*) and chromatin modifiers (*ASXL1*, *EZH2*). Epigenetic regulators are commonly affected by missense, nonsense and frameshift loss-of-function mutations that are usually located throughout the gene, sometimes affecting hotspots [55]. Mutations in DTA genes (*DNMT3A*, *TET2*, *ASXL1*) account for the majority of mutation-driven clonal hematopoiesis of indeterminate potential (CHIP), thus representing an early driver event for hematological neoplasia [49,56].

*TET2* mutations are very frequent in MDS/MPN (30–40%) and, more specifically, CMML (60–65%). In fact, Tet2 deficient mice develop myeloid neoplasia with a CMML-like phenotype [57]. *TET2* encodes an enzyme that catalyzes the conversion of the modified DNA base methylcytosine to 5-hydroxymethylcytosine which plays a role in normal myelopoiesis. Deleterious mutations disrupt this enzymatic activity favoring myeloid tumorigenesis [58]. *TET2* mutations are associated with advanced age, clonal hematopoiesis and normal karyotype, probably constituting early pathogenic mutations associated with the ageing of hematopoietic stem cells [59]. They are mutually exclusive with *IDH1/2* mutations and often coexist with 4q24 LOH or mutations in *SRSF2* or *EZH2*, especially in CMML [59,60].

*DNMT3A* is mutated in 10% of MDS/MPN, with a higher prevalence in patients with advanced age. *DNMT3A* mutations account for 50% of all clonal hematopoiesis mutations and are considered an early genetic event in disease-initiation process that confers a clonal advantage to the hematopoietic cells [61,62]. *DNMT3A* encodes the DNA methyltransferase responsible for the conversion of cytosine to 5-methylcytosine. Mutations result in an enzyme with reduced activity that commonly displays a dominant-negative effect [63]. They are associated with normal karyotype and are often co-mutated with *SF3B1* in patients with ring sideroblasts [4,64].

*ASXL1* disruptive mutations are seen in 40–50% of MDS/MPN and are especially prevalent in aCML (50–70%) and CMML (40–45%). *ASXL1* encodes a nuclear protein that plays a role in gene expression and chromatin remodeling, through the interaction with polycomb repressive complex 2 (PRC2) and transcription activators and repressors [65]. *ASXL1* mutations result in loss of epigenetic marks, promoting gene repression and myeloid transformation [66]. In MDS/MPN, they are frequently accompanied by other somatic mutations (*EZH2*, *PTPN11*, *SETBP1*, *SRSF2*, *STAG2, N/KRAS*) and they are associated with advanced disease features, such as leukocytosis or higher blast percentage [2,4,37].

*EZH2* mutations are detected in 15% of MDS/MPN and, more specifically, 25% of aCML. *EZH2* encodes the main catalytic subunit of the PRC2 complex, which has methyltransferase activity, and is considered to act as a tumor suppressor in myeloid malignancies [67]. The presence of loss-of-function mutations in *EZH2* is associated with deletions and LOH of chromosome 7q [67,68].

### 5.2. Splicing Factors

The spliceosome is a protein complex involved in the splicing process (intron removal) and the generation of a mature mRNA. Mutations in RNA-splicing machinery are associated with the presence of myelodysplasia, thus they are frequently detected in MDS and MDS/MPN and less commonly in AML and MPN [69]. Recurrent mutations have been reported in MDS/MPN in the spliceosome components *SF3B1*, *SRSF2*, *U2AF1* and *ZRSR2*, affecting overall up to 65% of cases, while mutations in *PRPF40B*, *SF3A1*, *SF1* and *U2AF2* are rare (<3%). Splicing factor mutations are mostly mutually exclusive missense heterozygous mutations localized at hotspot regions, that result in altered patterns of splicing that alter normal hematopoietic differentiation [69,70,71,72].

*SRSF2* mutations, mainly localized at hotspot P95, are seen in 30% of MDS/MPN and, more specifically, CMML (50%). They contribute to myelodysplasia by mutant-specific effects on exon recognition, altering *SRSF2* normal sequence-specific RNA binding activity and driving mis-splicing of key hematopoietic regulators [71]. They frequently co-occur with *TET2* mutations, especially in CMML, and they are associated with monocytosis and marked thrombocytopenia [2,4,69].

*SF3B1* mutations are detected overall in 25% of MDS/MPN, but they are highly specific of MDS/MPN-RS-T (90%) [26]. *SF3B1* mutations are associated with down-regulation of key gene networks involved in hematopoiesis, and most recurrent K700E mutation has been demonstrated to cause impaired erythropoiesis [72,73]. *SF3B1* mutations are a clear example of a genotype-phenotype correlation, since they are detected in up to 90% of patients with ring sideroblasts (MDS/MPN-RS-T and MDS-RS) [73,74].

*U2AF1* and *ZRSR2* mutations are seen in 8% and 4% of MDS/MPN, respectively. They result in aberrant pre-mRNA splicing and alter normal hematopoiesis [75,76].

### 5.3. Signaling Pathways

Signal transduction is a highly regulated process by which a signal is transmitted through a cell as a series of molecular events, most commonly protein phosphorylation catalyzed by protein kinases, which ultimately regulate cellular processes such as cell proliferation, apoptosis and differentiation. Signal transduction genes are generally affected by gain of function missense mutations in hotspot locus that constitutively activate the given signaling pathway. Up to 40% of MDS/MPN overlap syndromes harbor this type of mutation, which are usually associated with cytokine deregulation and inflammation. Mutations affecting signaling pathways are commonly acquired throughout disease evolution, constituting secondary events that, in many cases, drive disease progression to more advanced proliferative stages and ultimately AML [4,50].

Oncogenic RAS pathway is the most frequently affected signaling pathway in MDS/MPN, with recurrent mutations in *NRAS*, *KRAS*, *CBL*, *PTPN11* and *NF1*. Hyperactive RAS signaling is the main driving event in JMML, which is characterized by the presence of somatic mutations in *K/NRAS* and *PTPN11* in 50% of patients, and germline mutations in *CBL* and *NF1* [38]. RAS pathway mutations are associated with leukocytosis and extramedullary disease, and thus are also common in aCML and MP-CMML [4,30,51]. In contrast, they are very rare in MDS/MPN-RS-T (<3%), which is characterized by the presence of recurrent *JAK2* mutations (35%) that are associated with thrombocytosis and that constitutively activate the JAK/STAT signaling pathway [74]. Other genes, less frequently mutated in MDS/MPN (<5%), that also codify for kinases and other molecules involved in signal transduction, include *CSF3R*, *ETNK1*, *SH2B3* and *MPL* [4].

### 5.4. Transcription Factors

Transcription factors are proteins that bind to DNA-regulatory sequences (enhancers and silencers), usually localized in the 5′-upstream region of target genes, to modulate the rate of gene transcription. Approximately 30% of MDS/MPN patients harbor somatic loss-of-function mutations in a transcription factor gene, including *RUNX1*, *GATA2*, *CUX1*, *ETV6* and, less frequently (<3%), *NPM1*, *CEBPA* and *WT1*. As a somatic event, they can be present either in the ancestral clone or in subclonal populations and probably constitute driver events that occur after founder mutations in epigenetic regulators and splicing factors [4]. Of note, germline mutations in *RUNX1*, *GATA2* and *CEBPA* have been reported in myeloid neoplasms with germline predisposition [1].

Deleterious *RUNX1* mutations are seen in up to 15% of MDS/MPN. They are associated with thrombocytopenia and higher BM blast count [4,30,77] (Table 2).

### 5.5. Cohesin Components

The multiprotein cohesin complex is involved in the cohesion of sister chromatids and the post-replicative DNA repair, and it is codified by the genes *STAG1/2*, *SMC1A*, *SMC3* and *RAD21*. Loss-of-function nonsense and frameshift mutations in these genes, which are usually mutually exclusive, have been described in myeloid neoplasia [78]. 

*STAG2* is recurrently mutated in up to 8% of MDS/MPN, while mutations in other components of the cohesin complex are very rare (<3%) [4].

### 5.6. Other Functional Pathways

*SETBP1* mutations are found in 15% of MDS/MPN, with a higher prevalence in aCML (40%) [79,80]. *SETBP1* gene encodes a nuclear protein that inhibits tumor suppressor PP2A phosphatase activity through SET stabilization. Missense mutations in *SETBP1* avoid protein degradation, which constitutively inhibits PP2A, promoting an increase in cell proliferation [79]. These mutations usually correspond to secondary events and are associated with leukocytosis, −7/del(7q) and i(17)(q10) [30,60,81,82].

*TP53* mutations are, overall, rare in MDS/MPN (<5%), although this frequency increases in MDS/MPN-U (12%) and in therapy-related CMML [4,83]. *TP53* is a tumor suppressor gene involved in several functions, including DNA damage response, cell cycle arrest, apoptosis and cell senescence. Mutations in *TP53* are prevalent across different types of cancer and affect cell survival and proliferation. In myeloid neoplasms they are frequently associated with CK and commonly accompanied by a loss in the other allele [84,85].

## 6. Molecular Landscape of MDS/MPN and Clinical Implications

### 6.1. Chronic Myelomonocytic Leukemia

CMML is the most common and thus the best genetically characterized MDS/MPN, with >90% patients showing ≥1 mutation frequently affecting *TET2* (60%), *SRSF2* (50%) and *ASXL1* (45%) (Figure 1B). CMML is a disease of aging, and previous studies suggest that it represents the leukemic conversion of the myelomonocytic-lineage-biased aged hematopoietic system, in which mutated co-expression of *TET2* and *SRSF2* results in clonal hematopoiesis skewed toward monocytosis [52]. In fact, the combination of mutations in these two genes, as well as biallelic *TET2* mutations, is commonly present in the founder clone and is highly associated with CMML phenotype [2,4,32,53]. Recurrent secondary driver hits that contribute to clonal expansion and disease evolution include *RUNX1* (20%), *SETBP1* (12%) and *EZH2* (8%), which contribute to the MD-CMML phenotype, and *ASXL1*, seen in both MD an MP-CMML, being more prevalent in the latter [4,31,32,50,54]. In contrast, mutations in signaling genes, mainly RAS pathway (30%) and *JAK2* (10%) mutations, are associated with MP-CMML, characterized by leukocytosis, splenomegaly, constitutional symptoms, higher number of mutations and reduced survival. Finally, acquisition of RAS pathway mutations or chromosomal abnormalities frequently drive progression to AML, which is reported in 15–30% of CMML [8,30,33,51,54]. Similarly, mutations in AML-related genes, such as *NPM1* and *FLT3*, although very rare, can be acquired during the course of the disease and are highly suggestive of AML transformation [86,87] (Figure 3A).

According to the 2017 WHO Classification, the presence of mutations in genes often associated with CMML, which are detected in the vast majority of patients, can be used to support a CMML diagnosis in the proper clinical context [1]. This is especially useful in cases with mild dysplasia, given the high prevalence of normal karyotypes (up to 75%), and the fact that reactive sustained monocytosis is often seen in a spectrum of conditions such as chronic inflammation, bacterial (e.g., tuberculosis, listeriosis, bacterial endocarditis) and viral infections (e.g., EBV), post-splenectomy, and autoimmune diseases, which can be present in up to 20% of CMML patients, reinforcing the role of molecular testing. Moreover, the presence of CMML-associated genotypes (e.g., *TET2/SRSF2*) can be useful for the differential diagnosis between CMML and other myeloid-related neoplasms which can also present with monocytosis such as CML, MPN or M4 or M5 AML [4]. The inclusion of molecular markers in CMML-specific prognostic scoring systems has allowed to refine previous models. Current models combine classical adverse prognostic factors (age, anemia, thrombocytopenia, leukocytosis, presence of circulating immature myeloid cells or increased blast count) together with cytogenetic and molecular features [2,30,31]. More specifically, mutations in *ASXL1* have been consistently associated with unfavorable outcomes and thus are included in all these scores [2,30,31] (Table 2). In addition, molecular CMML-specific prognostic scoring system (CPSS-Mol) also incorporates mutations in *RUNX1*, *NRAS* and *SETBP1*, which are associated with inferior OS and higher risk of AML transformation [30]. In contrast, *TET2* mutations are associated with better outcomes, especially in the absence of *ASXL1* mutations [32,33,34]. Furthermore, these patients (TET2^MUT^/ASXL1^WT^) also show better response to therapy with hypomethylating agents (HMA) [32]. Of note, although HMA therapy can restore hematopoiesis in a subset of CMML patients, which is relevant in the context of cytopenias, it does not decrease mutation allele burden or prevent the acquisition of new genetic alterations, even in responders [88].

### 6.2. Atypical Chronic Myeloid Leukemia

Molecular profile of aCML is heterogeneous, with recurrent mutations in a wide number of genes. It is characterized by a high frequency of mutations in *ASXL1* (60–80%), which commonly constitute major founder initiating events [3,4]. Other recurrent mutations include *SETBP1* (40%), *SRSF2* (40%), *TET2* (35%), *EZH2* (25%), *NRAS* (20%), *RUNX1* (20%), *GATA2* (18%), *CBL* (15%); 10% for *CSF3R*, *STAG2* and *CUX1*; and a long tail of genes mutated in <10% of patients [3,4,35] (Figure 1B). Secondary hits that contribute to clonal expansion and cytopenia development are frequently seen in *SETBP1*, *RUNX1* and *EZH2*, while subclonal events that promote leukocytosis and drive eventual AML progression frequently affect RAS pathway genes (*N/KRAS*, *CBL*, *PTPN11* and *FLT3*) [3,4,35] (Figure 3B).

Although according to WHO criteria aCML can be differentiated from classic CML, CMML, chronic neutrophilic leukemia (CNL) and other MPN mainly on morphological basis, differential diagnosis of aCML is complex [1]. In this scenario, and in the right clinical and morphological context, molecular profiling can inform diagnosis, since specific gene mutation combinations seem to be highly associated with aCML (*ASXL1*/*SETBP1*, *SETBP1*/*SRSF2*, *ASXL1*/*EZH2* and *RUNX1*/*EZH2*) [4]. In contrast, mutations in *CSF3R* are highly specific of CNL (50–80%) and, if present in aCML (≤10%), they are frequently accompanied by *ASXL1* and/or *SETBP1* mutations [4,89]. Atypical CML is the most aggressive of all overlap syndromes, with a very poor survival and a high transformation rate to AML [90]. Prognostic impact of *SETBP1* mutations is controversial [4,35,79], while mutations in *TET2*, *RUNX1*, *NRAS* and *CUX1* have been reported to confer poor outcome [3,4]. Mayo Prognostic Model for aCML combines age, anemia and *TET2* mutations to stratify patients into two different risk groups that can allow the identification of patients who can benefit from HSCT, which is currently the hallmark of aCML treatment [3].

### 6.3. MDS/MPN with Ring Sideroblasts and Thrombocytosis

Compared to other overlap syndromes, MDS/MPN-RS-T is molecularly the least complex, with normal karyotypes in up to 85% of patients, lower number of mutations per patient and only a few genes involved in the pathogenesis of the disease [4]. Patients with MDS/MPN-RS-T have both morphological and molecular features of MDS-RS and MPN essential thrombocythemia (ET). This overlap syndrome is characterized by frequent mutations in *SF3B1* (90%), that strongly correlate with BM ring sideroblasts, and mutations in *JAK2* (40%) that correlate with thrombocytosis [26,74]. However, unlike in ET, *CALR* mutations are very rare (<3%) [4,36]. Other recurrent mutations include those associated with clonal hematopoiesis: *TET2* (20%), *ASXL1* (20%) and *DNMT3A* (15%); and less frequent mutations in *EZH2* (7%), *SETBP1* (5%) and *SRSF2* (5%) [4,15,26] (Figure 1B). *SF3B1* mutation is the major driver of the disease, and is recurrently preceded by *DNMT3A* mutations, that probably constitute a previous asymptomatic clonal hematopoiesis event [4]. Co-expression of *SF3B1* with *DNMT3A*, *TET2* or *ASXL1* promotes myelodysplasia with ring sideroblasts, while acquisition of *JAK2* mutations, and less frequently *MPL* (4%) and *SH2B3* (4%) mutations, constitute secondary events that promote thrombocytosis and contribute to the myeloproliferative phenotype [4,26,36] (Figure 3C).

MDS/MPN-RS-T has a mild course, with longer OS compared to other overlap syndromes and a low rate of AML transformation (<5%) [26]. Features such as presence of anemia, history of thrombosis and abnormal karyotype contribute to decreased OS [26,91]. Mutations in *ASXL1*, *SETBP1* and *EZH2* have also been associated with adverse outcomes [4,26]. Some of these variables are included in Mayo Prognostic Model for MDS/MPN-RS-T, which is the only disease-specific prognostic scoring system proposed, that has recently been validated [4,26].

### 6.4. MDS/MPN Unclassifiable

Molecular landscape of MDS/MPN-U is the most heterogeneous, which is expected given that this group includes a variety of poorly defined MDS/MPN that do not meet criteria for other well-defined subtypes, highlighting the complexity of categorization of overlap syndromes. Overall, the most frequent mutations include *ASXL1* (40%), *JAK2* (25%), *TET2* (25%), *SRSF2* (25%), *EZH2* (20%), *U2AF1* (16%) and *RUNX1* (15%) [19,27,37] (Figure 1B). One study recently reported that different molecular profiles could be identified in MDS/MPN-U, which recapitulated molecular signatures that are significantly associated with other MDS/MPN entities [4]. Thus, MDS/MPN-U cases could be further categorized into five molecular subtypes: ‘CMML-like’ (presence of biallelic *TET2*, *TET2*/*SRSF2* or *RUNX1*/*SRSF2*); ‘aCML-like’ (presence of any of these gene mutation combinations: *ASXL1*/*SETBP1*, *SETBP1*/*SRSF2*, *ASXL1*/*EZH2*, *RUNX1*/*EZH2*); ‘MDS/MPN-RS-T–like’ (*SF3B1* mutation, either alone or in combination with *DNMT3A* or *JAK2*; or *DNMT3A*/*JAK2*); ‘*TP53*’ (presence of *TP53* mutations) and ‘Other’ (Figure 4). Molecular subtypes of MDS/MPN-U displayed hematological BM and PB counts in accordance with their phenotypic group. For example, the ‘CMML-like’ group displayed increased monocyte count and included a few patients that could be classified as oligomonocytic CMML (monocyte count of 0.5 to <1 × 10^9^/L, ≥10% monocytes) [92]. Similarly, ‘MDS/MPN-RS-T-like’ patients displayed a median percentage of RS higher than the other groups and, clinically, behaved similarly to MDS/MPN-RS-T patients, suggesting that it might be worth considering taking the presence of *SF3B1* mutations into account in patients with 5–15% of ring sideroblasts, as in MDS-RS. Moreover, OS was significantly different and mimicked the outcome of the corresponding MDS/MPN counterpart, with the ‘MDS/MPN-RS-T-like’ category being associated with the highest median OS and the patients within ‘*TP53*’ group having the most unfavorable prognosis (Table 2). Of note, the adverse prognostic impact of *TP53* had previously been reported in these patients [27]. This genomics-based stratification system could potentially allow for inclusion of MDS/MPN-U patients in disease specific/appropriate clinical trials.

### 6.5. Juvenile Myelomonocytic Leukemia

JMML is currently considered a bona fide RASopathy. Even when clinical and hematological criteria must be evaluated for establishing a diagnosis of JMML, it has been described that around 90% of patients harbor molecular alterations in one of five RAS pathway genes (*PTPN11*, *NRAS*, *KRAS*, *NF1* or *CBL*), which define genetically and clinically distinct subtypes (Table 3 and Figure 5). These genetic aberrations activate the RAS/MAPK pathway and are mutually exclusive in most cases [1]. Subtypes with mutations in *PTPN11*, *NRAS* and *KRAS*, are characterized by heterozygous somatic gain of function mutations, while JMML harboring mutations in *NF1* or *CBL* are defined by germline RAS disease and acquired biallelic inactivation of respective genes in hematopoietic cells [22]. In approximately 10% of cases, none of these mutations can be detected and it has been reported that a few of these cases harbor *RRAS* (GTPase with 50% homology to the RAS proteins) activating somatic mutations, which give rise to an atypical form of this hematological disorder, rapidly progressing to AML [93].

**Table 3 cancers-13-02120-t003:** Genetic subtypes of JMML with most relevant characteristics.

JMML Subtype and Frequency (%)	Age of Onset (Years, Median)	Mutation (Type and Location)	Clinical Features	Prognosis and Treatment Implications
*PTPN11* (40%)	2.1	Somatic missense mutations affecting exons 3 and 13.	Acquisition of *NF1* haploinsufficiency is a frequent subclonal event.	Rapidly fatal unless allogenic HSCT can be successfully performed.
*NRAS* (18%)	1.2	Somatic missense mutations affecting exon 2.	• Subtype with the highest clinical diversity. • Clinically, patients are well and show a normal or slightly elevated HbF.	Although a considerable percentage relapse after HSCT, others survive in its absence and that of slowly regressing disease.
*KRAS* (14%)	0.9	Somatic missense mutations affecting exon 2. One half of cases present monosomy 7.	• Most children are diagnosed before the age of 1 year. • They often present with particularly severe disease.	Low relapse rate after allogeneic HSCT.
*CBL* (12–18%)	0.9	Germline mutations located throughout the linker and ring finger domain (intron 7, exons 8 and 9). Most patients have 11q isodisomy in hematopoietic cells.	Most children with *CBL* mutations have self-limiting disease with persistence of clonal hematopoiesis.	Observation without therapeutic intervention is generally advised. Value of allogenic HSCT is uncertain
*NF1* (5%)	2.8	≈65%: LOH at *NF1* locus caused by UPD of 17q ≈35%: compound heterozygous *NF1* inactivating mutations. Minority of cases: somatic interstitial deletions.	• Higher platelet count. • Higher percentage of bone marrow blasts.	Invariably fatal unless allogenic HSCT is successful.

Information adapted from Niemeyer, 2018 and Niemeyer and Flotho, 2019 [22,94]. Abbreviations: HbF: fetal hemoglobin. HSCT: hematopoietic stem cell transplantation. JMML: juvenile myelomonocytic leukemia. LOH: loss of heterozygosity. UPD: uniparental disomy.

In addition to the main RAS pathway mutation, secondary abnormalities are present in approximately half of the cases. In 10–15% of cases, these correspond to second hits in one of the RAS pathway genes (RAS double mutant cases) [95]. *SETBP1* mutations are found in 7–9% of the cases [39,95], followed by mutations affecting components of the PRC2 (*EZH2* and *ASXL1*), that are mutated in 4–7% of the cases [95,96,97]. Finally, *JAK3* mutations are also reported in 3–10% of JMML patients [39,95,97,98]. Secondary mutations are often subclonal and may be involved in disease progression rather than initiation of leukemia [95,97,99].

Unlike for adult MDS/MPN subtypes, it has been reported that in JMML distinct methylation signatures correlate with clinical and genetic features and are highly predictive of relapse following HSCT [38,40,100]. According to the methylation level, three groups that correlate molecular features and clinical outcome have been proposed: the high methylation group, characterized by somatic *PTPN11* mutations and poor clinical outcome; the intermediate methylation group, which shows enrichment for somatic *KRAS* mutations and monosomy 7; and the low methylation group, characterized by enriched for somatic *NRAS* and *CBL* mutations and a favorable prognosis [100].

## 7. Practical Consideration within the Clinical Context

It has been widely demonstrated within this review that genetic alterations play a role in the clinical heterogeneity of MDS/MPN neoplasms. NGS has revolutionized the field from every clinical point of view: diagnosis, prognosis and therapeutic management. In general, the finding of a given somatic mutation provides evidence of clonal hematopoiesis, which in the appropriate clinical context, and always complemented by traditional cytomorphological analyses, might guide patient’s diagnosis [1,49]. Molecular studies are also interesting in guiding treatment decisions, especially through prognostic evaluation. It has been suggested that prognostic scoring systems could be improved by incorporating information on molecular abnormalities, and in fact current CMML prognostic scoring systems already incorporate this type of information [2,30,31]. Moreover, pretransplant molecular mutation analysis can help to detect biomarkers in patients with MDS/MPN, which may be subsequently used as minimal residual disease markers after HSCT [101].

It is true that NGS is a relatively expensive technique compared to conventional ones, therefore, it seems essential to evaluate the economic impact of the use of this technology. However, if we consider those cases where NGS can potentially shed light on therapeutic decisions such as treatment intensity, HSCT decision or identification of candidates for clinical trials, the cost of such technique might be traduced into patient’s management benefit [102,103]. Although the approaches to financing health care are extremely diverse and are country specific, the increasingly widespread use of NGS would ultimately impact on lowering costs.

Given the high molecular heterogeneity in MDS/MPN, the new genotype-phenotype associations that are being found and those that are yet to be discovered invite us to consider machine learning as a future tool to help us integrate this large amount of information. It is not expected that sequencing or even artificial intelligence could replace conventional techniques nor clinicians and scientists’ expertise, but it would help us to integrate all the data and consequently to establish associations more quickly and with higher accuracy [104,105,106].

## 8. Conclusions

Our understanding of hematological malignancies is inevitably associated with the progress of molecular genetics. In MDS/MPN, the vast majority of patients are affected by gene mutations that have an impact on the pathophysiological features of the disease and play a role in their clinical heterogeneity. However, the spectrum of mutations is heterogeneous, the driver genes are not specific to MDS/MPN, and some of these mutations can be present in individuals with CHIP. Given the high percentage of individuals with this condition, a somatic mutation is not sufficient to diagnose a myeloid neoplasia and it should be interpreted in conjunction with clinical context. Therefore, the diagnosis of MDS/MPN syndromes remains heavily reliant on BM and PB morphology, and although there is no substitute for morphology or pathology, incorporation of molecular data has a potential role in the diagnosis of overlap syndromes. Despite that the spectrum of mutated genes is similar between the different MDS/MPN and even with other myeloid neoplasms, co-expression of specific gene mutations can be suggestive of a given MDS/MPN phenotype, as reviewed above, and can therefore inform diagnosis in the right clinical and morphological context.

MDS/MPN are clinically very heterogeneous. Even in each specific MDS/MPN entity, OS range is wide, and so risk stratification is necessary to identify patients with poor outcome, that are at risk of leukemic transformation and that might benefit from more intensive therapies. In this context, gene mutations play an important role, since they can add prognostic value to classical prognostic factors. For instance, *ASXL1* mutations are independently associated with a poor outcome in the spectrum of myeloid neoplasms, including a shorter OS and a higher risk of AML progression. In CMML, aCML and MDS/MPN-RS-T, disease-specific scoring systems have been developed which already include molecular data. In MDS/MPN-U, the most heterogeneous overlap syndrome, a prognostically-relevant molecular classification has been recently proposed. All these prognostic stratifications could inform appropriate clinical therapeutic strategies and appropriate need and timing for allogeneic HSCT. Unfortunately, only selected MDS/MPN patients may undergo HSCT due to their advanced age and/or comorbidities [7,107,108]. Furthermore, only a small proportion of MDS/MPN patients could benefit from targeted therapeutic options, since gene mutations involving genes that are targetable (e.g., *IDH1/2*, *FLT3*, *TP53*) only affect a small subset of patients (<5%). Therefore, therapeutic options in MDS/MPN are still limited, and newer drugs and treatment modalities are much needed for the correct clinical management of these patients.

## Figures and Tables

**Figure 1 cancers-13-02120-f001:**
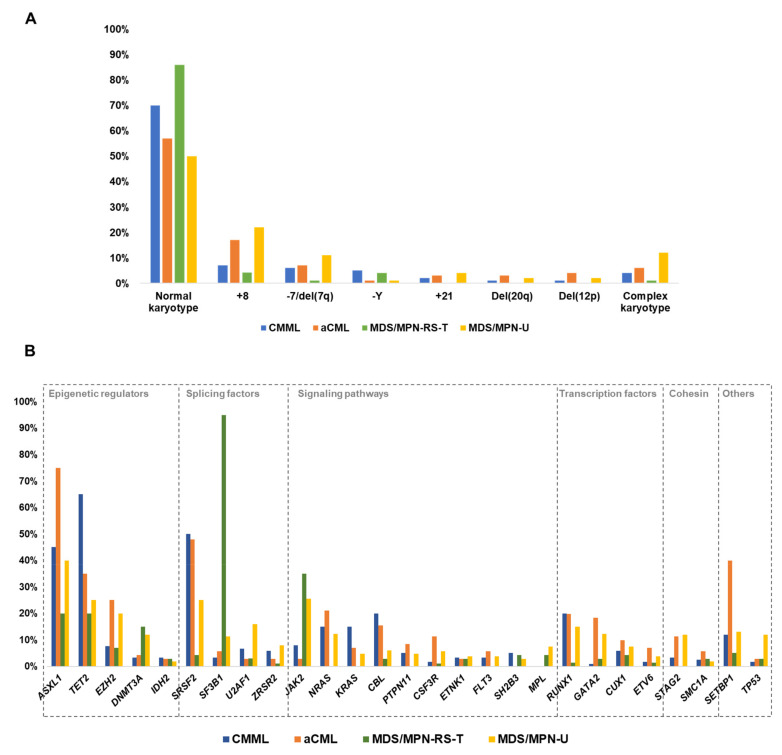
Genetic landscape of adult MDS/MPN. Frequency of recurrent cytogenetic alterations (**A**) and gene mutations (**B**) across the different subtypes of adult MDS/MPN. Based on data from Patnaik et al. [3], Palomo et al. [4], Breccia et al. [13], Jeromin et al. [15], DiNardo et al. [20], Such et al. [23], Tang et al. [24], Wassie et al. [25], Patnaik et al. [26] and Mangaonkar et al. [27]. Abbreviations: aCML: atypical chronic myeloid leukemia; CMML: chronic myelomonocytic leukemia; MDS/MPN-RS-T: myelodysplastic/myeloproliferative neoplasm with ring sideroblasts and thrombocytosis; MDS/MPN-U: myelodysplastic/myeloproliferative neoplasm unclassifiable.

**Figure 2 cancers-13-02120-f002:**
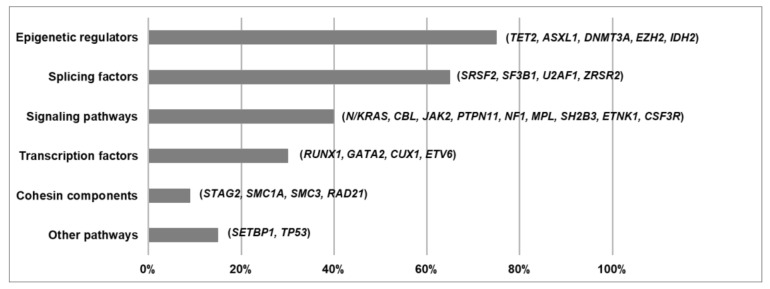
Functional pathways affected in MDS/MPN. Frequency of mutations affecting each pathway/functional category in MDS/MPN overlap syndromes. Based on data from Itzykson et al. [2], Patnaik et al. [3], Palomo et al. [4], Patnaik et al. [26] and Bose et al. [37].

**Figure 3 cancers-13-02120-f003:**
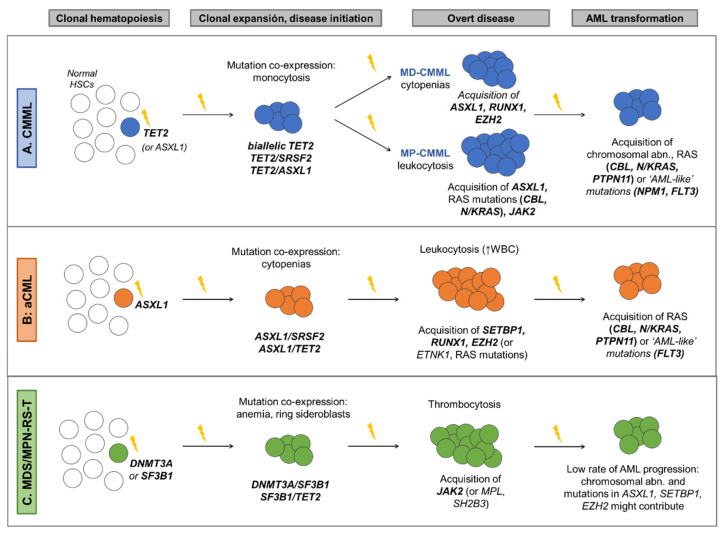
Clonal evolution model in adult MDS/MPN. The figure depicts the clonal evolution patterns most frequently observed in CMML (**A**), aCML (**B**) and MDS/MPN-RS-T (**C**). MDS/MPN arise from asymptomatic clonal hematopoiesis. Over time there is clonal expansion that leads to MDS/MPN phenotype and overt disease that, in some cases, eventually progresses to AML. This process takes place through the acquisition of molecular hits (chromosomal abnormalities and gene mutations) that confer to the neoplastic clone a selective advantage. The type of mutations and the order in which they are acquired shapes the disease phenotype and influences the clinical outcome. Based on data from Itzykson et al. [2], Palomo et al. [4], Elena et al. [30], Patnaik et al. [31], Coltro et al. [32], Palomo et al. [33], Meggendorfer et al. [35], Patnaik et al. [26], Steensma et al. [49], Itzykson et al. [50], Ricci et al. [51], Mason et al. [52], Awada et al. [53] and Patel et al. [54]. Abbreviations: aCML: atypical chronic myeloid leukemia; AML: acute myeloid leukemia; CMML: chronic myelomonocytic leukemia; HSCs: hematopoietic stem cells; MD-CMML: myelodysplastic CMML; MDS/MPN-RS-T: myelodysplastic/myeloproliferative neoplasm with ring sideroblasts and thrombocytosis; MDS/MPN-U: myelodysplastic/myeloproliferative neoplasm unclassifiable; MP-CMML: myeloproliferative CMML; WBC: white blood cell count.

**Figure 4 cancers-13-02120-f004:**
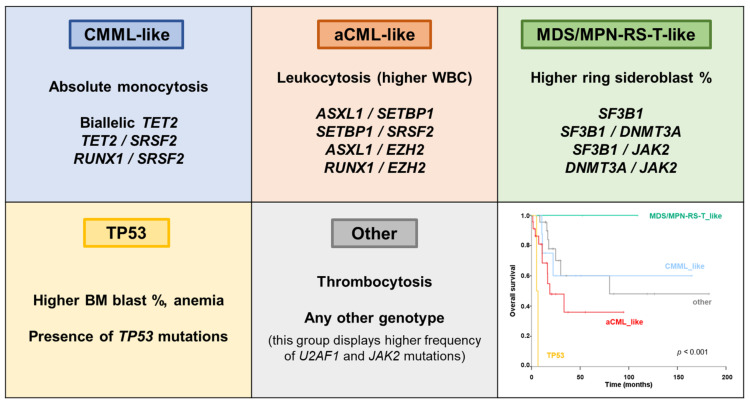
Molecular subtypes of MDS/MPN-U. Molecular classification of MDS/MPN-U based on the presence of specific gene mutation combinations. Morphological features of each subtype are also described in the corresponding panel. Overall survival of molecular subtypes is depicted at the bottom right corner. Based on data from Palomo et al. [4]. Abbreviations: aCML: atypical chronic myeloid leukemia; BM: bone marrow; CMML: chronic myelomonocytic leukemia; MDS/MPN-RS-T: myelodysplastic/myeloproliferative neoplasm with ring sideroblasts and thrombocytosis; WBC: white blood cell count.

**Figure 5 cancers-13-02120-f005:**
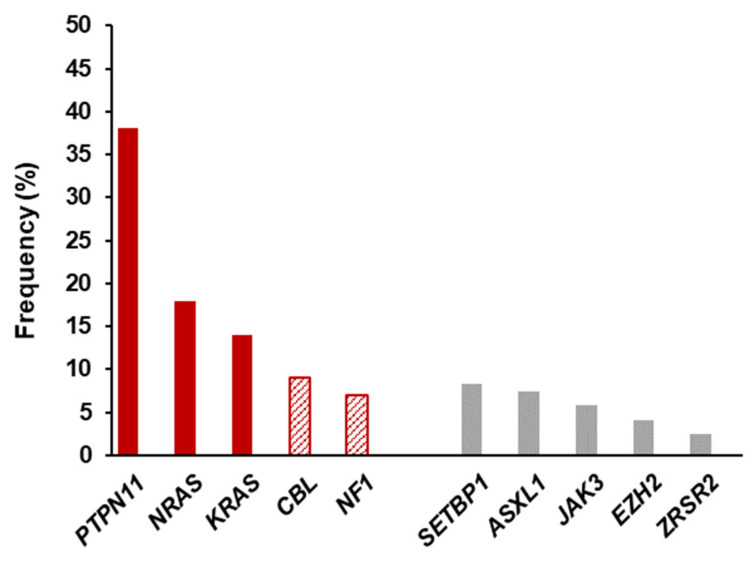
Molecular landscape of JMML. Frequency of recurrent gene mutations in JMML. Red bars represent RAS/MAPK pathway aberrations commonly found in 90% of the cases. Pattern filled bars represent common germline events. Grey bars depict recurrent subclonal secondary events in the disease. Based on data from Sakaguchi et al. [39], Stieglitz et al. [95], Sugimoto et al. [96], Caye et al. [97] and Bresolin et al. [98].

**Table 1 cancers-13-02120-t001:** Cytogenetic abnormalities in MDS/MPN.

MDS/MPN	Abnormal Karyotypes (%)	Common Abnormalities (Frequency %)
CMML	30%	+8: 6–7% −Y: 4–6% −7/del(7q): 2–9% +21: 1–2% CK: 3–6% Deletions of 20q (1–2%) and 12p (1%)
aCML	43%	+8: 17% −7/del(7q): 6–8% CK: 4–8%
MDS/MPN-RS-T	10–17%	+8: 4% −Y: 4% CK: 0–4%
MDS/MPN-U	50%	+8: 14–25% −7/del(7q): 11% CK: 12%
JMML	19–35%	−7: 9–25% Others (del(7q), +8): 10%

Abbreviations: aCML: atypical chronic myeloid leukemia; CMML: chronic myelomonocytic leukemia; CK: complex karyotype; JMML: juvenile myelomonocytic leukemia; MDS: myelodysplastic syndrome; MPN: myeloproliferative neoplasm; MDS/MPN-RS-T: myelodysplastic/myeloproliferative neoplasm with ring sideroblasts and thrombocytosis; MDS/MPN-U: myelodysplastic/myeloproliferative neoplasm unclassifiable.

**Table 2 cancers-13-02120-t002:** Clinical relevance of cytogenetic abnormalities and gene mutations in MDS/MPN.

MDS/MPN Subtype	Diagnosis	Prognosis
CMML	-WHO [1]: presence of mutations in genes often associated with CMML (*TET2, SRSF2, ASXL1, SETBP1*) in the proper clinical contest can be used to support diagnosis -Associated with the following gene mutation combinations: *TET2-SRSF2,* biallelic *TET2*, *SRSF2-RUNX1* [2,4,30]	Cytogenetics -Three cytogenetic stratification systems have been proposed [23,24,25] -Recurrent findings: • Low risk karyotypes: normal karyotype, isolated loss of Y • High risk karyotypes: chr7 abnormalities, complex karyotype, monosomal karyotype Gene mutations: -Unfavorable outcome: mutations in *ASXL1*, *RUNX1, NRAS* and *SETBP1* [2,30,31] -Favorable outcome: *TET2* mutations, especially in the absence of *ASXL1* mutations (*TET2^MUT^/ASXL1^WT^*). These patients also show better response to HMA [32,33,34]. Prognostic stratification: -GFM Model [2], stratification in 3 risk groups based on: Age > 65 years; Hb < 10 g/dL in females and <11 g/dL in males; WBC > 15 × 10^9^/L; Platelet count < 100 × 10^9^/L; *ASXL1* mutations -Mayo Molecular Model (MMM) [31], stratification in 4 risk groups based on: Hb < 10 g/dL; AMC > 10 × 10^9^/L; Platelet count < 100 × 10^9^/L; Presence of circulating IMCs; *ASXL1* mutations -CPSS-Mol [30], stratification in 4 risk groups based on: WBC ≥ 13 × 10^9^; BM blasts ≥ 5%; RBC transfusion dependency; Genetic risk group (includes CMML-specific cytogenetic risk stratification [23] and mutations in *ASXL1, RUNX1, NRAS* and *SETBP1*).
aCML	-Associated with the following gene mutation combinations: *ASXL1/SETBP1, SETBP1/SRSF2, ASXL1/EZH2, RUNX1/EZH2* [3,4,35]	Unfavorable outcome: mutations in *TET2, RUNX1, NRAS* and *CUX1* [3,4] Prognostic stratification: Mayo Prognostic Model for aCML [3], stratification in 2 risk groups based on: Age > 67 years; Hb < 10 g/dL; *TET2* mutations
MDS/MPN-RS-T	-WHO [1]: presence of a *SF3B1* mutation. -Associated with the following gene mutation combinations: *SF3B1*, either alone or in combination with *DNMT3A* or *JAK2*, or *DNMT3A/JAK2* [4,26,36]	Unfavorable outcome: -Presence of altered karyotype [4,26] -Mutations in *ASXL1, SETBP1, EZH2* [4,26] Prognostic stratification: Mayo Prognostic Model for MDS/MPN-RS-T [26], stratification in 3 risk groups based on: Hb < 10 g/dL; Abnormal karyotype; mutations in *ASXL1* or *SETBP1*
MDS/MPN-U	-	Unfavorable outcome: -Presence of chr7 abnormalities and complex karyotypes [19] -Mutations in *ASXL1, CBL, CEBPA, EZH2, STAG2, TP53* [4,27,37] Prognostic stratification: -Genomics-based stratification system (Figure 4), classification in 5 subtypes with prognostic relevance based on mutational profile [4]
JMML	-WHO [1]: presence of (1 finding sufficient): • Somatic mutation: *PTPN11, KRAS, NRAS* • Clinical diagnosis of *NF1* or *NF1* mutation • Germline *CBL* mutation CBL LOH	Prognostic stratification: According to the methylation level, three groups that correlate molecular features and clinical outcome have been proposed [38]: • High: characterized by somatic *PTPN11* mutations and poor clinical outcome • Intermediate: enriched in somatic *KRAS* mutations and monosomy 7 • Low: characterized by somatic *NRAS* and *CBL* mutations and a favorable prognosis

Abbreviations: aCML: atypical chronic myeloid leukemia; AMC: absolute monocyte count; chr: chromosome; CMML: chronic myelomonocytic leukemia; CPSS-Mol: molecular CMML-specific prognostic scoring system; GFM: Groupe Francophone des Myelodysplasies; Hb: Hemoglobin; HMA: hypomethylating agents; HSCT: hematopoietic stem cell transplantation; IMCs: immature myeloid cells; JMML: juvenile myelomonocytic leukemia; LOH: loss of heterozygosity; MDS/MPN-RS-T: myelodysplastic/myeloproliferative neoplasm with ring sideroblasts and thrombocytosis; MDS/MPN-U: myelodysplastic/myeloproliferative neoplasm unclassifiable; RBC: red blood cells; WBC: white blood cell count; WHO: World Health Organization.

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
