# Peer review of "Genetic Aspects of Myelodysplastic/Myeloproliferative Neoplasms"

_cancers, 2021, doi:10.3390/cancers13092120_

Round 1

Reviewer 1 Report

This is an excellent concise review of MDS/MPN disorders.  It could benefit from a critical analysis and how MDS/MPN is an overlap of MDS and MPN and its pathogenesis from a stem/progenitor cell. 

Reviewer 2 Report

Authors summarized the genetic findings across all MDS/MPN very well in this review article. 

However, there are other recent review articles discusses the very similar subject. Such as, "Molecular genetics of MDS/MPN overlap syndromes" by Hunter and Padron, 2020 and "Making sense of the MDS/MPN neoplasms overlap syndromes" by Tiu and Sekeres, 2014. Reader would like to see what is different/new in this manuscript that doesn't exist in other recent reviews. 

Reviewer 3 Report

The authors present a review on genetic aspects of MDS/MPN. The topic is of interest, there are some points to clarify

  1. The manuscript is unnecessarily long, the part of the diagnostic criteria from lines 52 to 99 can be abbreviated and referenced to their reference number 1 (WHO 2017).
  2. Table 1: likewise as before, this is unnecessary. Readers of this review certainly know the original publication and should be referenced to it.
  3. Sources for figure 1, 2, 3 and 5 should be referenced in the figure (following the same model the author used in the figure 4).
  4. In the abstract, starting line 26, the authors mentioned “Even when such alterations can be helpful for differential diagnosis, they should not be used alone as proof of neoplasia because some of these mutations may also occur in healthy older people”. The authors made this very appealing comment in the abstract, however throughout the review there is no in-depth elaboration of this concept except that mutations may be present in chip patients and in monocytosis.
  5. Monocytosis and finding of mutation, confirm the diagnosis of CMML? Actually a dilemma of clinical practice. Can the authors shed some light on this issue?
  6. Since for the different diseases they made a chapter of clinical implication and sometimes they referred to transplantation. In transplant patients, there is a way to select which of all the markers is the best for the follow-up of minimal residual disease. A comment on this in each entity would be interesting.
  7. Line 513: The authors said: “Unfortunately, HSCT is not an option for many MDS/MPN patients, given their advanced age”. Transplantation indication in MPN / MDS deserves a thoughtful comment, many centers perform transplants even in elderly patients, I would suggest formulating the comment with more caution and of course putting the bibliographic references.
  8. It would be interesting if the authors comment on some aspects, for example very aged patients, minimum basic gene battery to evaluate. What would essential and relevant in such type of patients, what would define the limit of doing everything or less.
  9. Real world data: it would be interesting to discuss how genetic evaluation of MDS/MPN is implemented in real world. A comment about of the overload of the health system doing all these genetic studies in all these patients (cost / benefit / toxicity of the health system outside of clinical studies) and discuss future perspective on this issue would be welcome.
  10. In the age of artificial intelligence, how the authors anticipated MDS/MPN algorithms in the future.

Round 2

Reviewer 2 Report

Authors pointed out my concern and explained well in their letter and did modifications to the review. 

Reviewer 3 Report

The authors have satisfactorily answered my questions.
I think the changes made have improved the quality of the manuscript.